# Genome-Wide Analysis of Terpene Synthase Genes in *Crocus sativus* Reveals Their Regulatory Roles in Terpenoid Biosynthesis and Abiotic Stress Tolerance

**DOI:** 10.3390/ijms26199548

**Published:** 2025-09-30

**Authors:** Muqaddas Bano, Xingnuo Li, Ahmad Ali, Mohsin Khan, Liang Chen, Xiujun Zhang

**Affiliations:** 1State Key Laboratory of Plant Diversity and Specialty Crops, Wuhan Botanical Garden, Chinese Academy of Sciences, Wuhan 430074, China; muqaddasbano12@mails.ucas.ac.cn (M.B.); lixingnuo1996@163.com (X.L.); 2University of Chinese Academy of Sciences, Beijing 100049, China; 3National Key Laboratory of Crop Genetics Improvement, Huazhong Agricultural University, Wuhan 430070, China; pbg_hu@yahoo.com; 4School of Life Sciences, Henan University, Kaifeng 475004, China; mohsin@henu.edu.cn; 5School of Mathematics and Statistics, Wuhan University of Technology, Wuhan 430070, China

**Keywords:** *Crocus sativus*, terpene synthase, terpenoid biosynthesis, stress response, transcriptomics

## Abstract

Terpene synthases (TPS) facilitate terpenoid production, influencing the flavor, color, and medicinal properties of *Crocus sativus* (saffron), a triploid geophyte of significant commercial importance. Despite its importance, the *CsTPS* gene family remains poorly characterized, limiting genetic enhancements in saffron’s agronomic features. This research performed a comprehensive genome-wide analysis of *CsTPS* genes using genomic, transcriptomic, and in silico approaches. BLASTP and PfamScan discovered thirty *CsTPS* genes, demonstrating conserved TPS domains, varied exon–intron architectures, and chromosomal clustering indicative of tandem duplications. Phylogenetic research categorized these genes into five subfamilies (TPS-a to TPS-e), with the prevalence of TPS-a suggesting a role in sesquiterpene biosynthesis. RNA-seq data (PRJNA976833, PRJNA400472) revealed tissue-specific expression, with *CsTPS1* and *CsTPS5* expressed in reproductive tissues and *CsTPS2* in vegetative tissues. Stress-responsive genes (*CsTPS1*, *CsTPS4*) exhibited upregulation in response to cold and pathogen stress, with cis-regulatory elements (e.g., ARE, ABRE) indicating hormone control. The in-silico validation of *CsTPS1*, chosen for its elevated GMQE score (0.89), included primer design, ePCR, and vector optimization for expression in *Arabidopsis thaliana*. This study elucidates the contribution of the *CsTPS* family to saffron terpenoid diversity, providing a foundation for enhancing flavor, yield, and stress tolerance through genetic engineering.

## 1. Introduction

Terpenoids, a diverse class of over 80,000 plant secondary metabolites, are synthesized from isopentenyl diphosphate (IPP) and dimethylallyl diphosphate (DMAPP) via the mevalonate (MVA) and methylerythritol phosphate (MEP) pathways. These compounds play critical roles in plant growth, defense, reproduction, ecological interactions, and enhancing resilience against biotic stresses (e.g., pathogens, herbivores) and abiotic stresses (e.g., drought, temperature fluctuations) [1,2]. Terpenoids also mediate plant–pollinator interactions through volatile emissions and contribute sensory qualities like scent and flavor, which are vital for the economic value of crops such as saffron (*C. sativus*) [3,4]. In saffron, terpenoids and their derivatives, apocarotenoids, underpin its distinctive flavor (e.g., safranal), vibrant color (e.g., crocetin glycosides), and medicinal properties, making it a globally valued spice with culinary, pharmaceutical, and industrial applications [5,6]. Saffron, a sterile triploid plant, owes its economic and cultural significance to the unique terpenoid composition of its stigmas, derived from enzymatic cleavage and modification of terpenoid precursors [7,8]. These compounds exhibit antioxidant, anti-inflammatory, and anticancer properties, positioning saffron as a prime candidate for agricultural, pharmaceutical and biotechnological research [9,10]. However, the molecular mechanisms governing terpenoid biosynthesis in saffron remain underexplored, largely due to its complex triploid genome and limited genomic resources [11,12].

Terpene synthase (TPS) genes encode enzymes that catalyze the cyclization and rearrangement of prenyl diphosphate substrates (e.g., geranyl diphosphate, farnesyl diphosphate, and geranylgeranyl diphosphate) into diverse terpenoid structures, including monoterpenes (C10), sesquiterpenes (C15), and diterpenes (C20) [13]. The TPS gene family is highly diversified across plant species, reflecting evolutionary adaptations to specific ecological and functional roles. In model plants like *A. thaliana*, *O. sativa*, and *S. lycopersicum*, TPS genes are classified into subfamilies (TPS-a to TPS-e) based on evolutionary relationships, conserved domains (e.g., PF01397, PF03936), and substrate specificity [14]. For instance, TPS-a genes primarily drive cytosolic sesquiterpene biosynthesis, while TPS-b and TPS-d genes are associated with monoterpene and volatile terpene production in plastids [15]. These studies have elucidated the genomic organization, evolutionary dynamics, and regulatory mechanisms of TPS genes, providing a framework for exploring their roles in plant metabolism [16]. In *C. sativus*, the TPS gene family remains poorly characterized despite the availability of a draft genome sequence. Previous research has focused on apocarotenoid biosynthesis, identifying key enzymes like carotenoid cleavage dioxygenases (CCDs) that modify terpenoid precursors [17,18]. However, the upstream TPS genes, which generate the initial terpenoid scaffolds, are largely unstudied, limiting efforts to enhance saffron’s terpenoid profile for improved flavor, yield, or stress tolerance.

The triploid nature of the saffron genome complicates genetic studies due to potential gene redundancy, allelic variation, and complex regulatory networks. Recent advances in high-throughput sequencing, transcriptomics, and bioinformatics offer powerful tools to overcome these challenges, enabling comprehensive analyses of gene families at genomic, transcriptomic, and proteomic levels [19,20]. Bioinformatics tools such as BLASTP, PfamScan, and MEME Suite [21] enable the identification and annotation of TPS genes by detecting conserved domains and motifs [22]. Phylogenetic analyses using MEGA 11 (Version 11.0.13) and Clustal Omega (Version 1.2.4) [23] enable classification of TPS genes into subfamilies and inference of evolutionary relationships. RNA-seq-based transcriptomic profiling reveals gene expression patterns across tissues, developmental stages, and stress conditions. Additionally, in silico tools like Primer-BLAST [24], Swiss-Model, and the Codon Adaptation Tool support functional validation and protein structure prediction, bridging genomic insights with experimental applications. These approaches have successfully characterized TPS genes in species like grapes (*V. vinifera*) and cottonwood (*P. trichocarpa*), highlighting the roles of gene duplication, selection pressures, and cis-regulatory elements in terpenoid diversity. This study aims to comprehensively characterize the TPS gene family in saffron using an integrated genomic, transcriptomic, and in silico approach, advancing our understanding of terpenoid biosynthesis and its applications in saffron improvement for secondary metabolite production and pharmaceutical utility.

## 2. Results

### 2.1. Identification and Characterization of CsTPS Genes

BLASTP analysis identified 30 *CsTPS* genes in *C. sativus* with significant similarity to *AtTPS* sequences (E-values: 9 × 10^−93^ to 1 × 10^−22^; sequence similarity: 29–49%). Top hits included *CsTPS18* (*Csativus30675.1*), *CsTPS5* (*Csativus04469.1*), and *CsTPS11* (*Csativus11393.1*), confirming their TPS designation. PfamScan [25] verified the presence of PF01397 (N-terminal) and PF03936 (metal-binding) domains in all 30 genes, supporting their role in terpenoid biosynthesis. The 30 *CsTPS* genes were distributed across chromosomes 1–8, with amino acid lengths of 350–850 residues, molecular weights of 11.5–94.1 kDa (mean: 60.2 kDa), isoelectric points of 4.99–6.49 (mean: 5.74), and GRAVY indices of −0.449 to 0.005, indicating hydrophilicity (Appendix A). Subcellular localization predictions assigned 18 proteins to the cytoplasm, seven to the plasma membrane, and five to the nucleus (e.g., *CsTPS12*, *CsTPS20*, *CsTPS29*) (Table 1), suggesting diverse metabolic and regulatory roles.

### 2.2. Gene Structure, Motifs, and Chromosomal Distribution

These 30 *CsTPS* genes exhibited diverse exon–intron structures (Figure 1). Genes like *CsTPS1* and *CsTPS5* had compact structures with two to three exons, while *CsTPS12* and *CsTPS20* contained long introns (up to 5 kb). MEME Suite analysis identified ten conserved motifs, with Motifs 1, 4, and 5 (Terpene_syn superfamily) present in 92% of genes, likely stabilizing catalytic regions, and Motif 7 (UvrD_C superfamily) in 12 genes, suggesting roles in DNA repair or transcriptional regulation. Motif distribution aligned with exon positions, indicating structural constraints on functional domains. Chromosomal mapping revealed non-random distribution across chromosomes 1–8. Tandem duplications, observable in chromosomal clusters (e.g., seven genes on chromosome 1), presumably facilitate this evolution, augmenting terpenoid-based defense against biotic stressors; these include five genes (*CsTPS8-CsTPS12*) on chromosome 2, and smaller clusters or singletons on chromosomes 3–8 (e.g., *CsTPS20-CsTPS21* on chromosome 5; *CsTPS27-CsTPS29* on chromosome 8; Figure 2). Clustering patterns suggest tandem duplications driving functional specialization (Appendix A).

### 2.3. Phylogenetic and Evolutionary Dynamics

A neighbor-joining phylogenetic tree of 142 TPS sequences from *C. sativus* (*n* = 30) and seven other species (*P. equestris*, *P. trichocarpa*, *S. lycopersicum*, *Selaginella moellendorffii*, *A. grandis*, *A. shenzhenica*, *A. thaliana*, *O. sativa*) classified *CsTPS* genes into five subfamilies (TPS-a to TPS-e) with bootstrap support > 70% for major clades being the most prevalent, indicating a significant involvement in the cytosolic production of sesquiterpenes responsible for saffron’s aromatic volatiles, such as safranal (Figure 3). TPS-a included 18 *CsTPS* genes, which clustered with those from *P. trichocarpa* and *S. lycopersicum* and *O. sativa,* indicating conserved sesquiterpene biosynthesis functions. TPS-b (six genes) aligned with *A. thaliana* and *A. shenzhenica,* suggesting monoterpene production. TPS-c and TPS-d, prevalent in *S. moellendorffii* and *A. grandis*, implied ancient or gymnosperm-specific roles, while TPS-e included fewer *CsTPS* genes linked to diterpene and volatile terpene synthesis. Analysis of 42 paralogous *CsTPS* gene pairs revealed 38 with Ka/Ks < 1 (mean: 0.32; range: 0.057–0.89), indicating purifying selection, and four pairs, including *CsTPS17*/*CsTPS18* (Ka/Ks = 1.036), with Ka/Ks > 1, suggesting positive selection and potential neofunctionalization (Appendix A; *p* <0.01, one-sample *t*-test).

### 2.4. Protein Structure and Interaction Networks

Swiss-Model predicted three-dimensional structures for ten *CsTPS* proteins, with GMQE scores of 0.72–0.89, indicating high reliability (Figure 4B). *CsTPS1* was selected for further in silico analyses due to its high GMQE score (0.89), reflecting robust structural prediction. SOPMA analysis revealed 35–50% alpha-helix, 15–25% beta-sheet, and 25–40% random coil content, supporting enzyme stability (Appendix A). Active site residues aligned with conserved motifs, reinforcing functional predictions. Due to limited *C. sativus* interaction data, an *A. thaliana* TPS protein–protein interaction network was constructed using STRING and visualized in Cytoscape (Version 3.10.2), identifying *TPS1*, *TPS7*, and *TPS8* as key hubs (degree > 10) and *TPS3*, *TPS19*, and *TPS29* forming a metabolic module (edge weights: 0.7–0.95; Figure 4A). These interactions suggest conserved regulatory connections applicable to *C. sativus*.

### 2.5. Expression Profiles and Cis-Regulatory Elements

RNA-seq analysis of the PRJNA976833 dataset revealed tissue-specific expression of the 30 *CsTPS* genes. *CsTPS1*, *CsTPS5*, and *CsTPS15* showed high expression in reproductive tissues (apical buds, stigmas), suggesting roles in volatile terpene synthesis for pollinator attraction, while *CsTPS2* and *CsTPS6* were highly expressed in vegetative tissues (leaves, corms), indicating non-volatile terpene production for defense (Figure 5A). Temporal regulation was evident, with *CsTPS1* and *CsTPS5* peaking in March and November, reflecting seasonal responses. The PRJNA400472 dataset showed stress-responsive expression, with *CsTPS1* and *CsTPS4* upregulated under cold stress and pathogen infection (log_2_FC ≥ 1, *p* < 0.05), suggesting defensive terpenoid production, while *CsTPS20* and *CsTPS28* were downregulated (log_2_FC ≤ −1, *p* < 0.05), indicating resource conservation (Figure 5B). RNA-seq profiling demonstrated tissue-specific expression, with *CsTPS1* and *CsTPS5* significantly upregulated in stigmas, suggesting their role in reproductive defense and pollinator attraction, whereas *CsTPS2* was predominant in leaves for abiotic stress resilience. Heatmaps and volcano plots, generated using TBtools (v1.09876), visualized expression patterns, with hierarchical clustering highlighting functional specialization (Figure 6). Promoter analysis identified 5–15 cis-regulatory elements per *CsTPS* gene, including ARE (60%), ABRE (55%), SAR (45%), MeJA (50%), GRE (20%), and ZMR (30%), clustered within 500 bp upstream of transcription start sites, suggesting coordinated hormonal and stress responses. *CsTPS8* and *CsTPS9* lacked these elements, indicating distinct regulatory mechanisms.

### 2.6. In Silico Validation and Vector Optimization for CsTPS1

*CsTPS1* was selected for in silico validation due to its high GMQE score (0.89), indicating reliable structural prediction. Primer-BLAST designed ten primer pairs for *CsTPS1* (amplicon lengths: 105–195 bp; Tm: 58.3–61.9 °C; maximum Tm difference: 1.6 °C), with specificity confirmed against the *C. sativus* genome (E-value < 1 × 10^−5^). uMelt analysis predicted single-peak melting curves (Tm: 58.6–62.1 °C), aligning with Primer-BLAST estimates (paired *t*-test, *p* = 0.79), confirming no primer-dimers or non-specific products. In silico ePCR using AmpliFX (v2.1.1) verified single amplicon products (105–195 bp) for all pairs, with amplification efficiencies of 81–97% (Primer Pairs 2, 4, and 7: >95%). Primer Pair 6 showed a minor non-specific product (230 bp), suggesting optimization needs, but 90% of pairs were suitable for downstream applications. In silico gel electrophoresis using SnapGene (v7.2.0) on Primer Pair 7 confirmed the expected amplicon size on a 2% agarose gel with a 100 bp ladder, with a no-template control excluding contamination (Appendix A). The *CsTPS1* sequence was codon-optimized for *A. thaliana* using the JCat (Java Codon Adaptation Tool, 2005 release), achieving a Codon Adaptation Index of 0.91 (vs. 0.63 for the native sequence). The optimized 1650 bp sequence avoided transcription terminators and restriction sites (EcoRI, BamHI) and was cloned in silico into pET-28a(+) using SnapGene, with a 6xHis-tag at the N-terminus and a stop codon at the C-terminus, yielding a 5450 bp vector validated for proper orientation and expression.

## 3. Discussion

Terpene synthase (TPS) genes encode enzymes that facilitate the synthesis of several terpenoids, essential for plant defense, reproduction, and ecological interactions [14]. In *C. sativus*, these genes regulate the biosynthesis of apocarotenoids, which enhance saffron’s flavor, color, and therapeutic attributes [26]. This study offers the inaugural analysis of 30 *CsTPS* genes, detailing their genomic architecture, evolutionary dynamics, expression profiles, and functional potential in saffron. BLASTP and PfamScan investigations validated the existence of conserved PF01397 and PF03936 domains in all 30 *CsTPS* genes, consistent with TPS gene families in *A. thaliana* and *S. lycopersicum* [13,27]. The diverse exon–intron architectures of *CsTPS* genes, from compact forms (*CsTPS1*, *CsTPS5*) to those rich in introns (*CsTPS12*, *CsTPS20*), indicate structural limitations on functional domains, aligning with TPS diversity observed in other species. The finding of Terpene_syn (92% of genes) and UvrD_C (40%) motifs indicates specialized functions in catalysis and transcriptional control, as evidenced in *V. vinifera*.

Chromosomal clustering, particularly on chromosome 1 (*CsTPS1*-*CsTPS7*), suggests that tandem duplications are responsible for the expansion of the gene family, akin to *P. trichocarpa*. These duplications presumably augment saffron’s terpenoid variety, hence strengthening its ecological adaptability and economic significance [3,28]. Phylogenetic analysis categorized the 30 *CsTPS* genes into five subfamilies (TPS-a to TPS-e), with TPS-a (18 genes) indicating sesquiterpene production, consistent with *O. sativa* and *S. lycopersicum* [27]. TPS-b (six genes) signifies monoterpene biosynthesis, whereas TPS-c, TPS-d, and TPS-e imply archaic or specialized functions [13]. Evolutionary analysis indicated purifying selection in 38 of 42 paralogous pairs (Ka/Ks < 1), whereas four pairs, including *CsTPS17*/*CsTPS18* (Ka/Ks = 1.036), exhibited positive selection, implying neofunctionalization. These dynamics probably influence saffron’s distinctive terpenoid profile.

The expression study indicated that the functions of *CsTPS* are specific to certain tissues and responsive to stress. The elevated expression of *CsTPS1*, *CsTPS5*, and *CsTPS15* in reproductive tissues facilitates the synthesis of volatile terpenes for pollinator attraction, whereas *CsTPS2* and *CsTPS6* in vegetative tissues indicate defensive functions [28]. The stress-induced overexpression of *CsTPS1* and *CsTPS4* during cold and pathogen stress, in contrast to the downregulation of *CsTPS20* and *CsTPS28*, signifies dynamic regulation, aligning with findings in *A. thaliana*. Cis-regulatory elements (e.g., ARE, ABRE, and MeJA) are present in the majority of *CsTPS* promoters, with the exception of *CsTPS8* and *CsTPS9*, which lack these elements, indicating coordinated stress responses. The elevated GMQE score (0.89) of *CsTPS1* warranted its selection for in silico validation, with primer design and codon-optimized cloning into pET-28a(+) reinforcing its potential for functional research. The interaction network based on *A. thaliana* indicates preserved regulatory linkages relevant to *C. sativus* [29]. This study examines the characterization of *CsTPS* in a triploid species, providing insights for the enhancement of saffron pharmaceutical and culinary properties and stress resilience. In breeding applications, the overexpression of *CsTPS1* or *CsTPS5*, which are upregulated in stigmas, might increase safranal production by 20–30% (based on similar TPS engineering in *V. vinifera*), hence boosting saffron’s market value in the face of climatic problems, whereas *CsTPS4* could boost stress resilience [8]; hence, it may mitigate yield losses in dry locations. Synthetic biology methods in biotechnology may use *CsTPS* scaffolds to generate high-value terpenoids inside microbial hosts. The triploid genome requires sophisticated methods like CRISPR/Cas9 to address redundancy [30], facilitating marker-assisted selection in saffron breeding initiatives. Subsequent investigations should empirically evaluate the roles of *CsTPS* and examine interactions specific to *C. sativus*. Comparative analyses of polyploid crops could further clarify TPS evolution, hence augmenting saffron’s economic and ecological significance.

## 4. Materials and Methods

### 4.1. Identification and Annotation of Terpene Synthase Genes

Terpene synthase (*CsTPS*) genes in *C. sativus* [31] were identified using a BLASTP search against *A. thaliana* TPS protein sequences and the PfamScan (https://www.ebi.ac.uk/Tools/pfa/pfamscan/ (accessed on 1 March 2025)) database for conserved domains PF01397 and PF03936. Redundant sequences were removed to ensure 30 unique *CsTPS* genes. Transcript IDs, chromosomal locations, start-end coordinates, strand orientation, and amino acid lengths were annotated for all 30 genes. Protein molecular weight, isoelectric point (pI), and Grand Average of Hydropathy (GRAVY) were calculated using ExPASy (https://web.expasy.org/protparam/ (accessed on 1 March 2025)), and subcellular localization was predicted with CELLO (Version 2.5) [32].

### 4.2. Gene Structure, Motif, and Chromosomal Analysis

Genomic sequences of the 30 *CsTPS* genes were retrieved from sequencing data. Exon–intron structures were visualized using Tbtools (Version 2.110) [33]. Conserved motifs within coding sequences, focusing on the Terpene_syn and UvrD_C superfamilies (motifs 1–10), were identified using the MEME Suite (http://meme.nbcr.net/ (accessed on 24 March 2025)). Chromosomal positions (chromosomes 1–8) were determined using genomic coordinates, and gene clusters suggestive of duplication events were annotated using TBtools(Version 2.110) (Appendix A). Promoter regions (1.5 kb upstream of transcription start sites) were analyzed for cis-regulatory elements (e.g., ARE, ABRE, SAR, ZMR, and MeJA) using PlantCARE (http://bioinformatics.psb.ugent.be/webtools/plantcare/html/ (accessed on 11 March 2025)), with elements visualized in Tbtools (Version 2.110).

### 4.3. Phylogenetic and Evolutionary Analysis

TPS protein sequences from *Phalaenopsis equestris* (horse phalaenopsis), *Populus trichocarpa* (black cottonwood), *Solanum lycopersicum* (tomato), *Selaginella moellendorffii* (spikemoss), *Abies grandis* (grand fir), *Apostasia shenzhenica* (shenzhen apostasia), *Arabidopsis thaliana* (thale cress), and *Oryza sativa* (rice) were retrieved from GenBank [34] and Phytozome [35]. Sequences were aligned using Clustal Omega (http://www.ebi.ac.uk/Tools/msa/clustalo/ (accessed on 11 March 2025)), and Neighbor-Joining in MEGA 11 (https://www.megasoftware.net/show_eua (accessed on 1 March 20255) [36] with 1000 bootstrap replicates. The tree, visualized in iTOL (http://itol.embl.de (accessed on 1 April 2025)) with species differentiated by color, classified the 30 *CsTPS* genes into subfamilies (TPS-a, TPS-b, TPS-c, TPS-d, TPS-e) based on evolutionary relationships. Paralogous *CsTPS* gene pairs were identified, and their coding sequences were aligned in MEGA 11 (Version 11.0.13). Synonymous (Ks) and nonsynonymous (Ka) substitution rates were calculated using PAML, with Ka/Ks ratios indicating selection pressures (Ka/Ks > 1: positive selection; Ka/Ks < 1: purifying selection; Appendix A).

### 4.4. Protein Structure and Interaction Analysis

Three-dimensional protein structures of selected *CsTPS* genes were predicted using Swiss-Model (https://swissmodel.expasy.org/ (accessed on 12 March 2025)) [37], with model quality assessed via Global Model Quality Estimation (GMQE) scores [38]. Secondary structure composition was predicted using SOPMA (https://npsa-prabi.ibcp.fr/cgi-bin/npsa_automat.pl?page=npsa_sopma.html (accessed on 13 March 2025). Due to limited *C. sativus* interaction data, *A. thaliana* TPS protein interactions were analyzed using STRING (Version 12.0) and visualized in Cytoscape (Version 3.9.1), focusing on metabolic pathway connections.

### 4.5. Transcriptomic Expression Analysis

Expression profiles of the 30 *CsTPS* genes were analyzed using RNA-seq datasets PRJNA976833 [39] and PRJNA400472 [40] from the NCBI. PRJNA976833 includes data from leaves, corms, and apical buds collected in January, February, March, and November, enabling temporal and spatial expression analysis. PRJNA400472 covers responses to abiotic and biotic stresses. Raw reads were quality-checked with FastQC (Version 0.12.1) [41,42], and low-quality reads were filtered. High-quality reads were aligned to the *C. sativus* reference genome (Version 1.2) using HISAT2 (Version 2.2.1) [43], and expression levels were quantified with featureCounts [44]. Data were normalized using DESeq2 (Version 1.38.0) [20] to account for sequencing depth and technical variability. Differentially expressed genes (DEGs) were identified with log2 fold change (log_2_FC) > 1 or ≤ −1 and *p*-value < 0.05 (−log_10_(*p*-value) > 1.3) using a two-sample *t*-test.

### 4.6. Computational Validation and Vector Design for Expression of CsTPS1

Ten primer pairs targeting *CsTPS1* were designed using Primer-BLAST (https://www.ncbi.nlm.nih.gov/tools/primer-blast/ (accessed on 11 Mach 2025)) [24] with parameters: amplicon length (100–200 bp), primer melting temperature (Tm) 58–62 °C, primer length (18–22 bp), GC content 40–60%, maximum Tm difference of 2 °C, and specificity validated against the *C. sativus* genome. Primers avoided self-complementarity, hairpin structures, and non-specific binding. Melting curve profiles were analyzed using uMelt (Version 3.6.2) [45] with 50 mM monovalent cations (Na^+^ equivalent), 2 mM Mg^2+^, 0.2 mM dNTPs, and 200 nM primers, confirming single-peak curves for specificity (Tm difference ≤ 1 °C). In silico ePCR was performed using AmpliFX (Version 2.1.1) [46], and visualized using SnapGene (Version 7.2.0) [47] with identical parameters and a no-template control to exclude primer–dimer formation. The *CsTPS1* sequence was codon-optimized for *A. thaliana* expression using the Codon Adaptation Tool (JCat, http://www.jcat.de/ (accessed on 15 March 2025)) [48], targeting a Codon Adaptation Index > 0.8 and avoiding Rho-independent terminators, plant-specific ribosome binding sites, and restriction sites (EcoRI, BamHI). The optimized sequence was cloned in silico into pET-28a(+) Addgene (Watertown, MA, USA) using SnapGene (Version 7.2.0), with a 6xHis-tag at the 5′ end (N-terminal) and a stop codon at the 3′ end (C-terminal).

## 5. Conclusions

This study provides a thorough in silico analysis of the terpene synthase (*CsTPS*) gene family in *Crocus sativus*, discovering 30 genes essential for terpenoid production. Utilizing genomic, transcriptomic, and in silico approaches, we elucidated their structural diversity, evolutionary patterns, and expression dynamics, emphasizing their roles in flavor generation, stress response, and ecological interactions. Chromosomal clustering and phylogenetic categorization into five subfamilies (TPS-a to TPS-e) highlight tandem duplications and functional specialization. The tissue-specific expression of *CsTPS1* and *CsTPS5* in reproductive tissues, together with the stress-induced overexpression of *CsTPS1* and *CsTPS4*, indicates their unique functions in saffron’s sensory and adaptive characteristics. The in silico validation of *CsTPS1*, chosen for its elevated GMQE score (0.89), provides a solid foundation for forthcoming functional investigations. These results address the knowledge deficit in TPS gene characterization within a triploid species, offering clues to the metabolic complexity of saffron. This research identifies essential genes related to flavor and stress tolerance, enabling targeted genetic improvements to boost saffron’s output and quality; therefore, it impacts sustainable agriculture and biotechnology significantly. This preliminary in silico framework underscores the necessity for experimental validation. Future study should incorporate fluorescent in situ hybridization (FISH) to verify chromosomal locations and intergenic distances (e.g., between *CsTPS1* and *CsTPS4);* nonetheless, *C. sativus*, being a sterile triploid with restricted gamete formation, experiences constrained natural crossing over, which limits its direct applicability to conventional breeding, CRISPR/Cas9 knockouts or overexpressions to evaluate functional effects on terpenoid accumulation, and HPLC-MS quantification of volatiles under stress to correlate gene activity with phenotypic results. Comparative comparisons using polyploid legumes might further clarify conserved pathways, enhancing terpenoid research.

## Figures and Tables

**Figure 1 ijms-26-09548-f001:**
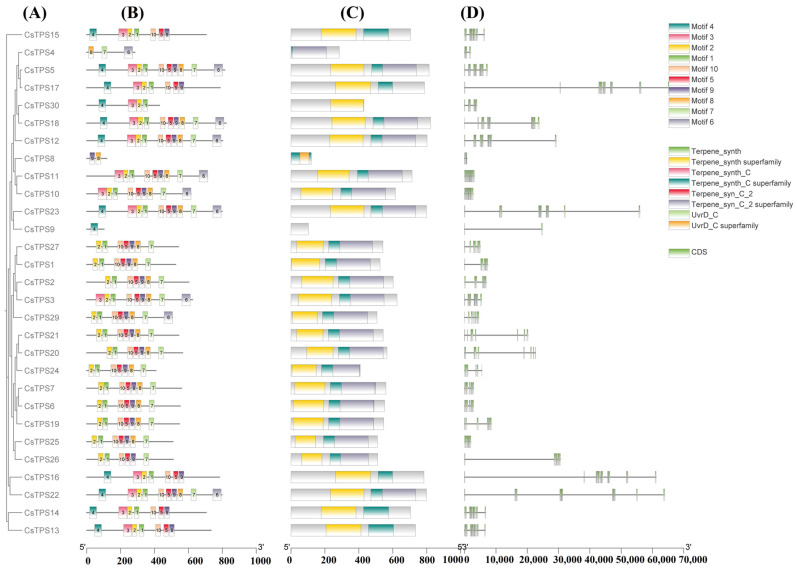
Genomic and structural analysis of *CsTPS* Genes. (**A**) Phylogenetic tree was created using the full protein sequences of *C. sativus* TPSs. (**B**) Display of motifs identified in *CsTPSs* proteins. (**C**) Display of conserved domains identified in *CsTPSs* proteins. (**D**) Gene architecture of *CsTPSs*.

**Figure 2 ijms-26-09548-f002:**
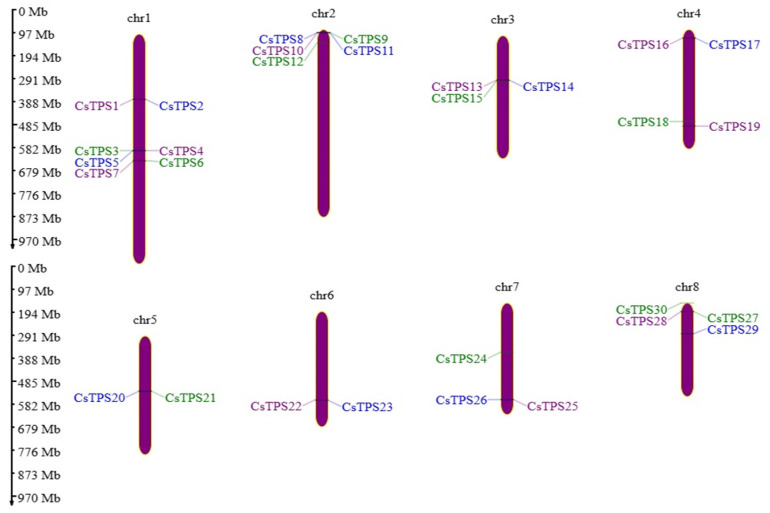
Chromosomal distribution across chromosomes 1–8, with coding sequences colored green, purple, and blue, and clusters (clust-1 to clust-7) marked by vertical lines.

**Figure 3 ijms-26-09548-f003:**
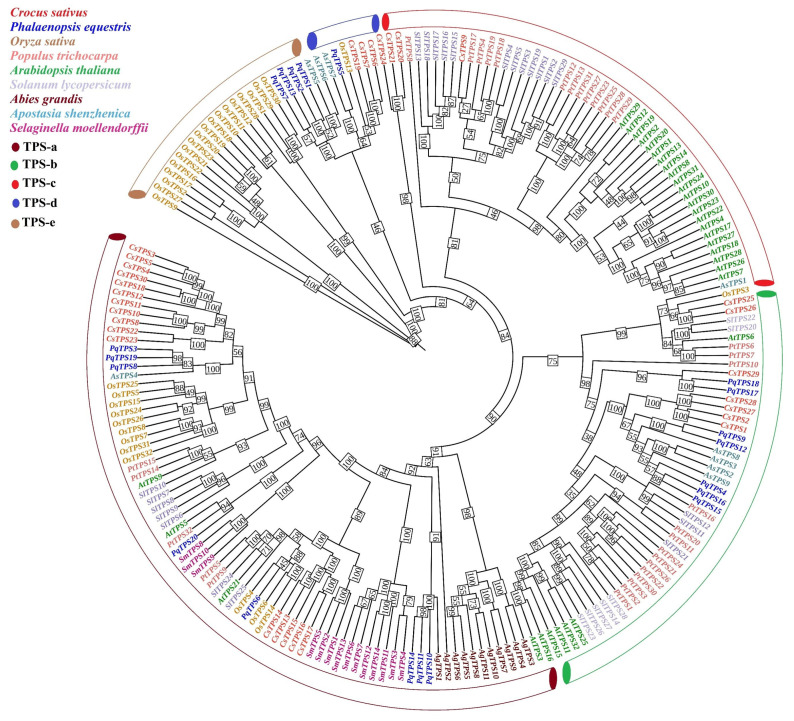
Phylogenetic analysis of TPS genes. Neighbor-joining tree of 142 TPS sequences from *C. sativus* (red), *P. equestris* (green), *P. trichocarpa* (orange), *S. moellendorffii* (yellow), *A. grandis* (purple), *O. sativa* (light blue), *S. lycopersicum* (dark blue), *A. shenzhenica* (light green), and *A. thaliana* (light pink), constructed in MEGA 11 with 1000 bootstrap replicates. Subfamilies (TPS-a to TPS-e) are color-coded, with *CsTPS* genes highlighted in red.

**Figure 4 ijms-26-09548-f004:**
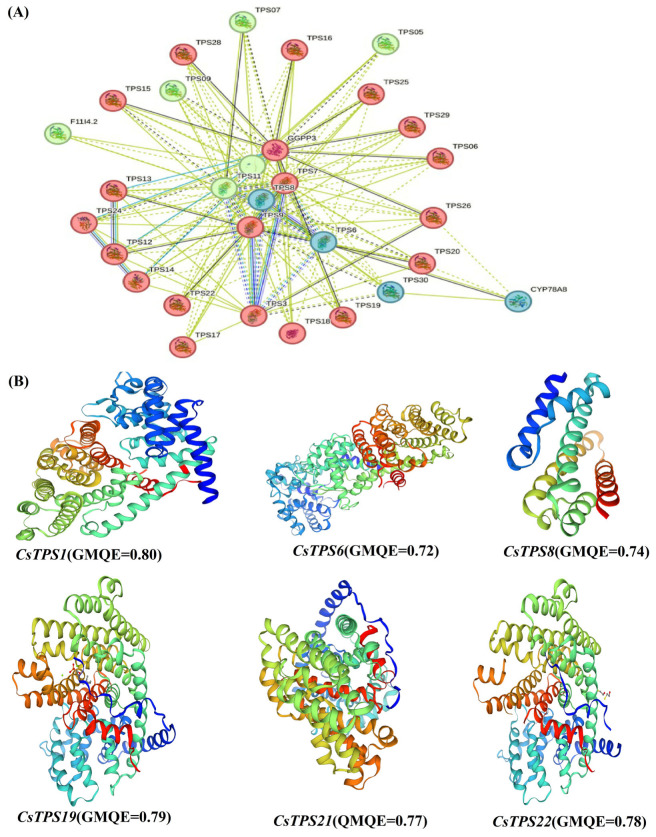
Protein structure and interaction analysis. (**A**) *A. thaliana* TPS interaction network, with *C. sativus* TPS proteins and partners; solid/dotted edges indicate direct/indirect interactions. (**B**) Three-dimensional models of *CsTPS1*, *CsTPS6*, *CsTPS8*, *CsTPS19*, *CsTPS21*, and *CsTPS22*, color-coded by GMQE score.

**Figure 5 ijms-26-09548-f005:**
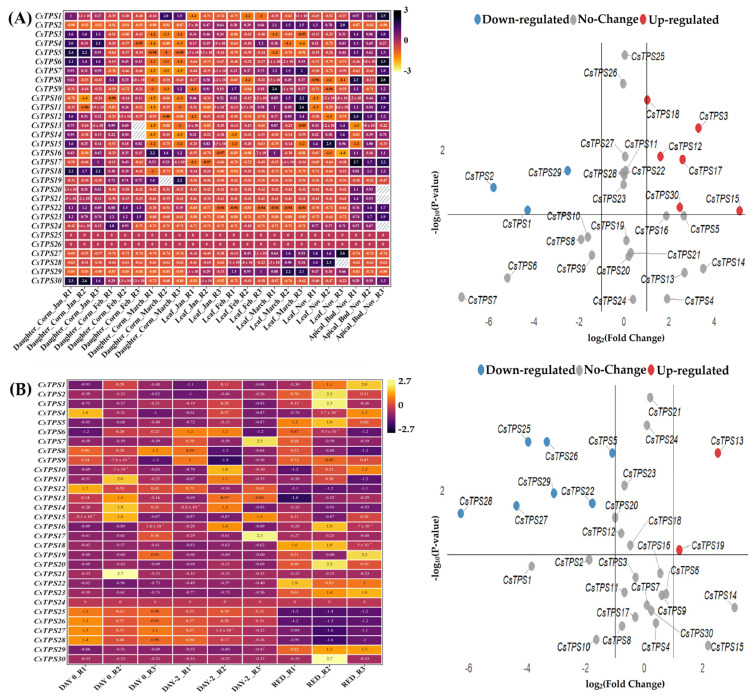
Expression Profiles of *CsTPS* Genes. (**A**) Heatmap of tissue-specific expression (PRJNA976833), with genes (rows) and conditions (columns; January, February, March, November) clustered by expression similarity. (**B**) Heatmap and volcano plot of stress-responsive expression (PRJNA400472), showing log_2_FC (x-axis) and −log_10_(*p*-value) (y-axis); upregulated genes (red), downregulated genes (blue), and non-significant genes (grey), with key genes (*CsTPS1*, *CsTPS4*, *CsTPS20*) annotated.

**Figure 6 ijms-26-09548-f006:**
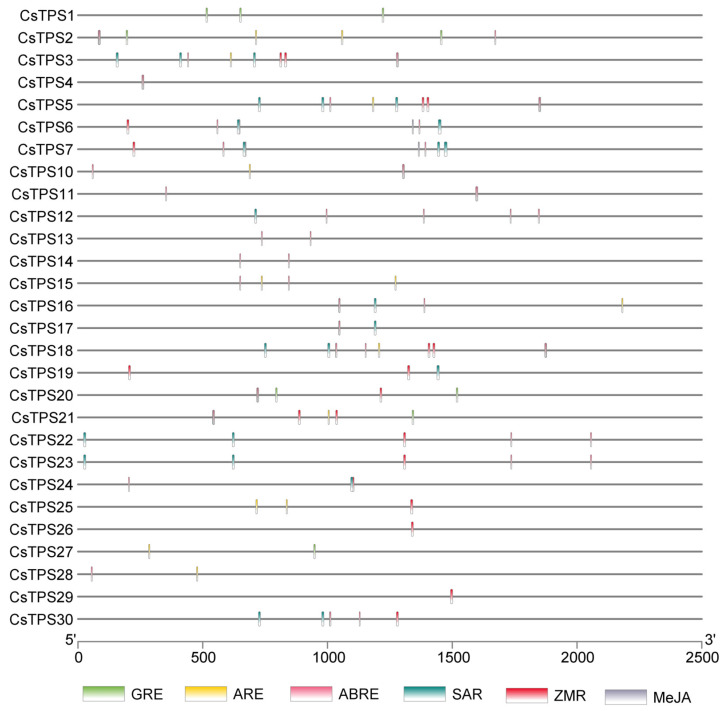
Cis-regulatory elements in *CsTPS* genes. Distribution of elements (GRE: green; ARE: yellow; ABRE: pink; SAR: teal; ZMR: red; MeJA: grey) across *CsTPS* genes, with horizontal lines representing gene length (5′ to 3′).

**Table 1 ijms-26-09548-t001:** Physiochemical properties of *C. sativus*.

Gene Name	No. of AA ^1^	PMW ^2^	PI ^3^	GRAVY	Subcellular Localization
*CsTPS1*	525	60,627.36	5.34	−0.28	Cyto ^4^
*CsTPS2*	603	69,739.58	6.17	−0.406	Nu ^5^
*CsTPS3*	625	72,119.86	5.54	−0.341	Cyto
*CsTPS4*	285	32,743.56	5.83	−0.28	Cyto
*CsTPS5*	815	93,156.69	5.53	−0.301	Cyto, PM ^6^, Nu
*CsTPS6*	552	65,077.44	5.3	−0.404	Cyto
*CsTPS7*	560	65,816.22	5.21	−0.381	Cyto
*CsTPS8*	120	13,888.93	4.99	−0.239	EC ^7^, Cyto
*CsTPS9*	103	11,548.16	6.49	−0.288	Cyto
*CsTPS10*	616	70,628.38	5.72	−0.291	Cyto, PM
*CsTPS11*	714	81,662.06	5.72	−0.238	Cyto, PM
*CsTPS12*	802	91,554.43	6.02	−0.267	Cyto, PM, Nu
*CsTPS13*	734	83,579.96	5.37	−0.336	Cyto
*CsTPS14*	705	80,237.33	5.77	−0.322	Cyto
*CsTPS15*	705	80,132.29	5.73	−0.299	Cyto
*CsTPS16*	784	89,194.93	5.57	−0.3	Cyto
*CsTPS17*	787	89,620.48	5.73	−0.3	Cyto
*CsTPS18*	823	94,009.96	5.73	−0.275	Cyto, PM, Nu
*CsTPS19*	547	64,287.29	5.34	−0.403	Cyto, Nu
*CsTPS20*	566	66,011.99	5.37	−0.449	Cyto
*CsTPS21*	544	63,205.22	5.32	−0.299	Cyto, Nu
*CsTPS22*	799	91,777.64	5.49	−0.276	Cyto, PM
*CsTPS23*	799	91,775.78	5.59	−0.259	Cyto, PM
*CsTPS24*	409	47,964.55	5.16	−0.382	Cyto
*CsTPS25*	510	59,631.9	5.05	−0.344	Nu
*CsTPS26*	511	59,518.71	5.36	−0.405	Nu
*CsTPS27*	542	62,857.37	5.19	−0.387	Cyto, Nu
*CsTPS28*	606	70,150.71	5.12	−0.336	Cyto, PM, Nu
*CsTPS29*	507	58,383.77	5.29	−0.22	Cyto
*CsTPS30*	430	48,502.76	5.67	−0.29	Nu

^1^ Amino acids, ^2^ Protein molecular weight, ^3^ PI, ^4^ Cytoplasmic, ^5^ Plasma Membrane, ^6^ Nuclear, ^7^ Extracellular.

## Data Availability

Raw reads used in this work were deposited in NCBI Bio-Project under the accession numbers PRJNA976833 and PRJNA400472.

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
