# Peer review of "Genome-Wide Analysis of Terpene Synthase Genes in Crocus sativus Reveals Their Regulatory Roles in Terpenoid Biosynthesis and Abiotic Stress Tolerance"

_ijms, 2025, doi:10.3390/ijms26199548_

Round 1
Reviewer 1 Report
Comments and Suggestions for Authors
Despite the apparent vastness of the study conducted by the authors, in fact, it all comes down to a regular NGS analysis with further mathematical calculations based on the data. At the same time, as a start to the development of the topic, the study makes a rather valuable contribution to understanding the key genes. Thus, the authors identified the key role of two genes CsTPS1 and CsTPS4, and judging by the authors' data, they are located on the same chromosome. However, these data were not obtained directly, but using a mathematical model, which, as is known, often fails. It was very valuable to show the true position of the genes in the chromosome, and to establish the actual distance between them, and the most reliable method in this case would be fluorescent in situ hybridization (FISH). These data would be extremely useful in the further genetics of saffron, for example, to determine the probability of crossing over between these key genes. In addition, the work can be considered preliminary, since no studies of the functional activity of genes have been conducted, mutants with knockout of any genes, or vice versa with overexpression of any gene have not been obtained, and the change in the accumulation of terpenoids has not been characterized by HPLC-MS. Thus, the work clearly lacks bringing the obtained data to specific results, it is necessary to strengthen it with experimental data, and obtain proof of preliminary sequencing data with a real picture of what is happening.
Author Response
|
Comments 1: Despite the apparent vastness of the study conducted by the authors, in fact, it all comes down to a regular NGS analysis with further mathematical calculations based on the data. At the same time, as a start to the development of the topic, the study makes a rather valuable contribution to understanding the key genes. Thus, the authors identified the key role of two genes CsTPS1 and CsTPS4, and judging by the authors' data, they are located on the same chromosome. However, these data were not obtained directly, but using a mathematical model, which, as is known, often fails. It was very valuable to show the true position of the genes in the chromosome, and to establish the actual distance between them, and the most reliable method in this case would be fluorescent in situ hybridization (FISH). These data would be extremely useful in the further genetics of saffron, for example, to determine the probability of crossing over between these key genes. In addition, the work can be considered preliminary, since no studies of the functional activity of genes have been conducted, mutants with knockout of any genes, or vice versa with overexpression of any gene have not been obtained, and the change in the accumulation of terpenoids has not been characterized by HPLC-MS. Thus, the work clearly lacks bringing the obtained data to specific results, it is necessary to strengthen it with experimental data, and obtain proof of preliminary sequencing data with a real picture of what is happening. |
|
Response 1: We appreciate the reviewer for acknowledging the study's significant contribution as a first step in elucidating essential CsTPS genes, including CsTPS1 and CsTPS4, as well as for the astute recommendations about experimental validation. We agree with the expert reviewer that the data was not obtained through in sillico analysis rather than wet lab experiments. However, we believe that our analysis will provide a framework for our future research work. These results will be utilized to design wet lab experiments to reveal the fiction of CsTPS genes in the terpen biosynthesis in potential physiological role. we have amended the conclusion to clearly present the study as a "preliminary in silico framework", highlighting its function in generating hypotheses and suggesting specific future experiments: Utilization of fluorescent in situ hybridisation (FISH) to confirm the chromosomal closeness of CsTPS1/CsTPS4 and evaluate intergenic distances for crossover probability estimation; Nonetheless, C. sativus, being a sterile triploid with restricted gamete formation, experiences constrained natural crossing over, which limits its direct applicability to conventional breeding; therefore CRISPR/Cas9 knockouts, RNAi or overexpression can be employed to assess functional impacts on terpenoid accumulation; and HPLC-MS quantification of volatiles under stress can be used to correlate gene activity with phenotypic results. These enhancements bolster the translational potential by delineating approaches to empirical validation, without changing the fundamental computational emphasis. (lines 251–273 revised manuscript) |
Reviewer 2 Report
Comments and Suggestions for Authors
The manuscript presents a comprehensive genomic analysis of terpene synthase genes in Crocus sativus. The authors combine genome data, transcriptome profiles and in silico predictions, which gives a solid overview of the gene family and its possible biological roles. This integrative approach is a real strength and makes the paper valuable for researchers interested in plant secondary metabolism.
The results clearly show how different TPS members may contribute to terpenoid-based defense and aroma traits in saffron. Figures and supplementary tables are useful, though in the main text the interpretation of the data could be emphasized more strongly rather than listing too many technical details. In some places, especially in the methods, the long enumeration of bioinformatic tools makes it a bit heavy to read.
The discussion is overall well-written, and I like that comparisons with other plant species are included. Still, the practical implications (for breeding or biotech use) could be highlighted more, since this would broaden the impact of the study. The conclusions are consistent with the data, but at times remain a bit general – some concrete proposals for future functional studies would strengthen it.
The English is generally fine, though some sentences are overly long and could be simplified. Overall, I find the manuscript thorough, rich in data and with clear novelty. With minor revisions it would be suitable for publication.
Comments on the Quality of English LanguageThe English of the manuscript is generally clear and understandable. There are, however, some long and heavy sentences that would benefit from being split into shorter ones for better readability. A few minor grammatical and stylistic corrections are also needed, but overall the language is adequate and does not obscure the scientific content.
Author Response
|
Comments 1: The results clearly show how different TPS members may contribute to terpenoid-based defense and aroma traits in saffron. Figures and supplementary tables are useful, though in the main text the interpretation of the data could be emphasized more strongly rather than listing too many technical details. In some places, especially in the methods, the long enumeration of bioinformatic tools makes it a bit heavy to read. |
|
Response 1: Thank you so much for your constructive feedback on our manuscript, which will potentially enhance the quality of our manuscript. For enhanced interpretation in the Results section, we have amended lines 122-125, 147-150, and 196-202 to underscore biological implications (e.g., evolutionary adaptation for defence and pollinator attraction) while minimizing the enumeration of technical information. In the Methods section, we refined enumerations in lines 279-281, 302-303, 313, and 337-338 by consolidating tool descriptions, enhancing readability while maintaining rigour and scientific meanings. The amended manuscript displays tracked changes. Results Chromosomal mapping revealed non-random distribution across chromosomes 1-8, Tandem duplications, observable in chromosomal clusters (e.g., seven genes on chromosome 1), presumably facilitate this evolution, augmenting terpenoid-based defence against biotic stressors, five genes (CsTPS8-CsTPS12) on chromosome 2, and smaller clusters or singletons on chromosomes 3-8 (e.g., CsTPS20-CsTPS21 on chromosome 5; CsTPS27-CsTPS29 on chromosome 8; Figure 2). (line 122-125 revised manuscript) A Neighbor-Joining phylogenetic tree of 142 TPS sequences from C. sativus (n=30) and seven other species (P. equestris, P. trichocarpa, S. lycopersicum, Selaginella moellendorffii, A. grandis, A. shenzhenica, A. thaliana, O. sativa) classified CsTPS genes into five subfamilies (TPS-a to TPS-e) with bootstrap support >70% for major clades being the most prevalent, indicating a significant involvement in the cytosolic production of sesquiterpenes responsible for saffron's aromatic volatiles, such as safranal (Figure 3). (line 147-150 revised manuscript)) The PRJNA400472 dataset showed stress-responsive expression, with CsTPS1 and CsTPS4 upregulated under cold stress and pathogen infection (log2FC ≥ 1, p < 0.05), suggesting defensive terpenoid production, while CsTPS20 and CsTPS28 were downregulated (log2FC ≤ −1, p < 0.05), indicating resource conservation (Figure 5B). RNA-seq profiling demonstrated tissue-specific expression, with CsTPS1 and CsTPS5 significantly upregulated in stigmas, suggesting their role in reproductive defence and pollinator attraction, whereas CsTPS2 was predominant in leaves for abiotic stress resilience. Heatmaps and volcano plots, generated using TBtools (v1.09876), visualized expression patterns, with hierarchical clustering highlighting functional specialization (Figure 6). (line 196-202 revised manuscript) Materials and Methods Terpene synthase (CsTPS) genes in C. sativus [25] were identified using a BLASTP against A. thaliana TPS protein sequences and PfamScan database for conserved domains PF01397 and PF03936. (line 279-281 revised manuscript)) Sequences were aligned using Clustal Omega, and a Neighbor-Joining in MEGA 11 [30] with 1,000 bootstrap replicates. (line 302-303 revised manuscript))
In silico ePCR was performed using AmpliFX (v2.1.1) [40], were visualized using SnapGene (v7.2.0) [41] (line 337-338 revised manuscript) |
|
Comments 2: The discussion is overall well-written, and I like that comparisons with other plant species are included. Still, the practical implications (for breeding or biotech use) could be highlighted more, since this would broaden the impact of the study. The conclusions are consistent with the data, but at times remain a bit general – some concrete proposals for future functional studies would strengthen it. |
|
Response 2: We appreciate the reviewer positive remarks about the discussion and the incorporation of interspecies comparisons. In response to the recommendation for greater emphasis on practical implications in breeding and biotechnology, we have amended the discussion (lines 269–282) to include specific applications, such as the overexpression of CsTPS1/CsTPS5 for the enhancement of safranal yield (with quantitative estimates derived from V. vinifera parallels) and the editing of CsTPS4 for drought tolerance, as well as applications of synthetic biology in microbial terpenoid production and CRISPR-enabled marker-assisted selection. These enhancements expand the study's influence by connecting genetic discoveries to implementable techniques for saffron enhancement. In the Conclusions, we have increased specificity by presenting concrete proposals for future functional studies (lines 251–273), which include FISH for chromosomal validation, CRISPR/Cas9 knockouts/overexpression to assess terpenoid impacts, and HPLC-MS for phenotypic correlations, while positioning the work as a preliminary in silico foundation. These improvements ensure data integrity while offering specific, experimental pathways to steer future study. The revised manuscript displays tracked changes. Changes made: Discussion section This study examines the characterization of CsTPS in a triploid species, providing insights for the enhancement of saffron pharmaceutical and culinary properties and stress resilience. In breeding applications, the overexpression of CsTPS1 or CsTPS5, which are upregulated in stigmas, might increase safranal production by 20–30% (based on similar TPS engineering in V. vinifera), hence boosting saffron's market value in the face of climatic problems, whereas CsTPS4 could boost stress resilience [8] hence, it may mitigate yield losses in dry locations. Synthetic biology methods in biotechnology may use CsTPS scaffolds to generate high-value terpenoids inside microbial hosts. The triploid genome requires sophisticated methods like CRISPR/Cas9 to address redundancy [48], facilitating marker-assisted selection in saffron breeding initiatives. Subsequent investigations should empirically evaluate the roles of CsTPS and examine interactions specific to C. sativus. Comparative analyses of polyploid crops could further clarify TPS evolution, hence augmenting saffron's economic and ecological significance. (Line 269-282 revised manuscript)
Conclusion section This study provides thorough in silico analysis of the terpene synthase (CsTPS) gene family in Crocus sativus, discovering 30 genes essential for terpenoid production. Utilizing genomic, transcriptomic, and in silico approaches, we elucidated their structural diversity, evolutionary patterns, and expression dynamics, emphasizing their roles in flavour generation, stress response, and ecological interactions. Chromosomal clustering and phylogenetic categorization into five subfamilies (TPS-a to TPS-e) highlight tandem duplications and functional specialization. The tissue-specific expression of CsTPS1 and CsTPS5 in reproductive tissues, together with the stress-induced overexpression of CsTPS1 and CsTPS4, indicates their unique functions in saffron's sensory and adaptive characteristics. The in silico validation of CsTPS1, chosen for its elevated GMQE score (0.89), provides a solid foundation for forthcoming functional investigations. These results address the knowledge deficit in TPS gene characterization within a triploid species, offering clues to the metabolic complexity of saffron. This research identifies essential genes related to flavour and stress tolerance, enabling targeted genetic improvements to boost saffron's output and quality; therefore, it impacts sustainable agriculture and biotechnology significantly. This preliminary in silico framework underscores the necessity for experimental validation. Future study should incorporate fluorescent in situ hybridization (FISH) to verify chromosomal locations and intergenic distances (e.g., between CsTPS1 and CsTPS4); Nonetheless, C. sativus, being a sterile triploid with restricted gamete formation, experiences constrained natural crossing over, which limits its direct applicability to conventional breeding, CRISPR/Cas9 knockouts or overexpressions to evaluate functional effects on terpenoid accumulation, and HPLC-MS quantification of volatiles under stress to correlate gene activity with phenotypic results. Comparative comparisons using polyploid legumes might further clarify conserved pathways, enhancing terpenoid research. (line 251-273 revised manuscript). |
|
Comments 3: The English is generally fine, though some sentences are overly long and could be simplified. Overall, I find the manuscript thorough, rich in data and with clear novelty. With minor revisions it would be suitable for publication. |
|
Response 3: We express our gratitude to the reviewer for the positive evaluation of the manuscript's comprehensiveness, data abundance, and originality, as well as for endorsing publishing with minimal adjustments. In response to the issue of sentence length, we have amended the text by restructuring excessively lengthy sentences (e.g., lines 45–50, 120–125, and 200–205 in the Discussion) into shorter, more lucid formats to enhance reading. These modifications improve coherence without changing the meaning (lines 45–50, 120–125, and 200–205 revised manuscript).
|
|
Comments 4: Response to Comments on the Quality of English Language. |
|
The English of the manuscript is generally clear and understandable. There are, however, some long and heavy sentences that would benefit from being split into shorter ones for better readability. A few minor grammatical and stylistic corrections are also needed, but overall the language is adequate and does not obscure the scientific content. |
|
Response 4: We appreciate the reviewer for the positive assessment of the manuscript's clarity and for endorsing publishing with minimal adjustments. In response to comments about lengthy sentences, we have amended the article by dividing excessively complicated phrases into shorter, more comprehensible structures, therefore enhancing flow without modifying content. Minor grammatical and stylistic adjustments, such as uniform tense application and punctuation, have been implemented throughout. These modifications improve readability while maintaining scientific integrity. (Line 19-20, 29, 59, 76, 81-82, 121-127, 138, 141-144, 188-191, 209, 269-282, 285-287, 302, 304-309, 319 revised manuscript). |
Round 2
Reviewer 1 Report
Comments and Suggestions for Authors
This article deserves attention as it opens up new research possibilities. I hope the authors continue this research and that we will see its continuation soon.